# The Complexity of the Human–Animal Bond: Empathy, Attachment and Anthropomorphism in Human–Animal Relationships and Animal Hoarding

**DOI:** 10.3390/ani12202835

**Published:** 2022-10-19

**Authors:** Emanuela Prato-Previde, Elisa Basso Ricci, Elisa Silvia Colombo

**Affiliations:** 1Department of Pathophysiology and Transplantation, University of Milan, 20133 Milan, Italy; 2Associazione Asino Anch’io, 20080 Basiglio, Italy; 3Petlife, 20151 Milan, Italy

**Keywords:** human–animal relationship, human–animal bond, empathy, attachment, anthropomorphism, animal hoarding

## Abstract

**Simple Summary:**

The relationship between humans and animals may have positive effects for both parties, but there are situations in which it has poor or even negative effects for animals or for both humans and animals. Several studies reported the positive effects of this relationship in which both humans and animals obtain physical and psychological benefits from living together in a reciprocated interaction. There is also clear evidence that human–animal relationships may be characterized by different forms and levels of discomfort and suffering for animals and, in some cases, also for people. This work depicts the complex and multifaceted nature of the human–animal relationship; describes the role of empathy, attachment and anthropomorphism in the human–animal bond; shows how these psychological processes are involved in a dysfunctional way in animal hoarding, with highly detrimental effects on animal well-being.

**Abstract:**

The human–animal relationship is ancient, complex and multifaceted. It may have either positive effects on humans and animals or poor or even negative and detrimental effects on animals or both humans and animals. A large body of literature has investigated the beneficial effects of this relationship in which both human and animals appear to gain physical and psychological benefits from living together in a reciprocated interaction. However, analyzing the literature with a different perspective it clearly emerges that not rarely are human–animal relationships characterized by different forms and levels of discomfort and suffering for animals and, in some cases, also for people. The negative physical and psychological consequences on animals’ well-being may be very nuanced and concealed, but there are situations in which the negative consequences are clear and striking, as in the case of animal violence, abuse or neglect. Empathy, attachment and anthropomorphism are human psychological mechanisms that are considered relevant for positive and healthy relationships with animals, but when dysfunctional or pathological determine physical or psychological suffering, or both, in animals as occurs in animal hoarding. The current work reviews some of the literature on the multifaceted nature of the human–animal relationship; describes the key role of empathy, attachment and anthropomorphism in human–animal relationships; seeks to depict how these psychological processes are distorted and dysfunctional in animal hoarding, with highly detrimental effects on both animal and human well-being.

## 1. Introduction

Since very ancient times, humans’ social world has comprised not only other humans but also different nonhuman species with whom humans have established relationships varying in form and strength. The relationship with animals has played a key role in our survival and evolution, and our way of considering them and relating to them has changed over the course of time, taking on different forms [1,2,3,4,5].

Scientific evidence from different disciplines, including psychology, sociology and animal welfare, shows that the relationship between humans and animals is complex, multifaceted, ambivalent and even paradoxical, with different consequences for animals and humans [6,7,8,9,10,11].

For more than 40 years, studies have shown that the human–animal relationship, especially that with domestic and companion animals, is the result of a complex interactions between biological, psychological, social and cultural factors [12,13,14,15,16,17].

The current work reviews some of the literature on the complex nature of the human–animal relationship to outline how it can range from highly positive to highly negative. Then, the role of empathy, attachment and anthropomorphism is described, aimed at highlighting their key role in determining the quality of human–animal relationships and bonds. Finally, we focus on animal hoarding, a highly dysfunctional relationship with animals, analyzing how empathy, attachment and anthropomorphism are involved in this psychological disorder in a dysfunctional or pathological way that leads to animal abuse and suffering.

## 2. The Multifaceted Nature of the Human–Animal Relationship

Comparative studies have revealed a wealth of commonalities between humans and nonhuman animals that allows them to engage in interspecific social relationships [14,18,19,20,21,22,23].

As pointed out in [14], one reason why humans are both willing and capable to relate to animals is the presence of basic biological structures and mechanisms that are relevant in social contexts; these mechanisms are shared between humans and other animals and are highly conserved among vertebrates. Basic mechanisms that enable human intraspecific close relationships and bonds appear to also be involved in our relationships and bonds with animals, and various nonhuman species may form intense and durable bonds with people (e.g., [24,25]).

Studies on companion and farm animals show that the way in which animals are considered, treated and cared for is strongly affected by peoples’ characteristics such as personality, attitudes, empathy and attachment levels and beliefs in animals’ mental capacities (e.g., [26,27]). In addition, sociodemographic variables, such as gender, age, family structure, education level and previous experiences with animals, play a role [8,12,13,28,29,30].

Gender differences are well documented in the literature, with women consistently showing higher levels of empathy and concern regarding animal suffering, holding more positive attitudes towards animals and being more engaged in animal protection and less prone to animal exploitation, animal abuse and cruelty [13,29,31,32,33,34]. Apostol et al. [13], for example, reported that gender was a good predictor of attitudes towards animals, together with empathy towards animals, anthropomorphic beliefs and owning a companion animal.

Personality has been associated with positive attitudes to animals, and there is a relationship between psychopathic personality traits and animal abuse and violence [26,35,36]. Empathy and attachment are both related to the quality of human–animal relationships and problems in empathy, attachment and emotion regulation are associated with animal abuse and cruelty [37,38,39,40].

Attitudes, broadly defined as psychological tendencies to evaluate a particular entity (e.g., humans or animals) with some degree of favor or disfavor [41], are important in shaping the human–animal relationship and bond and are reported to play a key role in determining animals’ health and welfare. According to [12], two main aspects underlie human attitudes toward animals: “affect”, which can be defined as people’s affective and emotional responses to animals, and “utility”, i.e., people’s perceptions of animals’ instrumental value. Serpell [12] suggests that the relative strength of these factors would depend on individual characteristics, experience, cultural factors and also on the specific attributes of animals.

The role of experience and culture in human–animal relationships is well documented in the literature: on the one hand the range of animals kept as pets around the world is exceptionally wide [42]; on the other hand, human–companion animal relationship styles vary considerably across cultures [7,16]. In Western countries, for example, dogs and cats are mainly kept for companionship, and the idea of eating them is considered intolerable and morally unacceptable; however, in other countries (e.g., China and Vietnam), these species are both kept as pets and consumed [43,44,45]. Moreover, in many Western countries, cats and dogs are the most popular and beloved companion animals, but, at the same time, a huge number of them (and other pets) are abandoned, neglected, abused and needlessly euthanized every year [46,47,48]. Finally, most people care about pet welfare, but for various reasons, there is still less concern regarding farm animals’ (e.g., pigs, cows and chickens) welfare [49,50].

Regarding animal characteristics, people generally do not see all animals as equal, as their physical and behavioral traits play a role in how they are perceived, considered and treated [12,17,51]. Humans tend to prefer animals that are phylogenetically close to them and perceived as physically, behaviorally or cognitively similar; these aspects trigger more positive affect and attachment and caregiving behaviors, as well as greater empathy and a higher concern in terms of welfare and conservation [52,53,54,55,56,57]. At the same time, in all human societies, animals are ranked on a ‘‘ladder of worth” as is almost everything else, including other humans [58,59].

Knight et al. [51] suggested that the variability in people’s attitudes to the use and exploitation of animals depends on a combination of different factors including beliefs about the mental capacities of animals, perceived superiority of humans, availability of alternatives to the use of animals for various purposes (e.g., medical research and food) and whether the problem of animal exploitation has any direct personal relevance. Belief in the animal mind appears to be a good predictor of attitudes towards animals and their use and abuse for the benefit of humans (e.g., entertainment, experimentation and financial gain [60,61]). The propensity to use animals is greater when people believe there is no alternative, when their knowledge of how animals are used is poor, when the affinity with animals is low and when the perceived benefits of using animals outweigh the costs [51].

Finally, there is evidence that people, when facing situations of conflict regarding animal well-being and suffering, tend to “build arguments” that justify and corroborate their existing attitudes or behavior to avoid dissonance [62]. This has been clearly demonstrated in the “meat paradox”, which shows that those that consume meat may overcome the cognitive dissonance resulting from a positive attitude towards both animals and meat by living either in a state of “tacit denial” regarding animals being killed to produce meat or by denying that animals can suffer [63,64]. In other words, mental abilities and the capacity for suffering tend to be attributed by people to animals when it is in their interest and motivation and not when it does not suit them [65].

## 3. Human–Animal Relationships: Two Sides of the Story?

Given the array of factors involved in most human–animal relationships, it is not surprising that people relate to animals in very different ways that range from highly positive, affectionate and caring to highly negative, dysfunctional or abusive [7,43,58,59,66,67]. As Ascione and Shapiro [68] pointed out, for every study showing that the human–animal relationship can be beneficial and built on love and caring, another deals with animal exploitation or abuse including the abandonment and neglect of companion animals or cases of dog fighting or animal hoarding. Moreover, attachment, empathy and concern for animals do not necessarily guarantee their welfare, and people may disagree on the proper way to treat animals or on what constitutes a fair human–animal relationship [8].

For example, Mota-Rojas et al. [69] outlined how adverse consequences on pets’ welfare might depend on widespread and apparently affectionate and caring behaviors, such as dressing pets, application of cosmetics, letting them sleep in beds or overfeeding them, emphasizing that people’s behavior towards companion animals should be based on the understanding and respect of their natural needs rather than on supposed similarities and an affective involvement.

Therefore, people’s relationships with animals in general (farm, zoo and wild animals) and with companion animals cannot be easily categorized into positive (i.e., caring and affectionate) and negative (i.e., neglecting or abusive).

A large body of scientific literature has considered the physical and psychological benefits that both humans and animals may obtain by living together in a reciprocated interaction (e.g., humans: 21, [70,71,72,73,74]; see [75,76] for critical analyses; animals: [77,78,79]). However, there is also extensive literature showing that human–animal relationships are characterized by different forms and levels of discomfort and suffering for animals (e.g., [11,68,69,80,81,82]).

As occurs in interpersonal relationships, human beings not only and not always care about the well-being of animals but also pursue their own personal needs, desires and goals; human’s psychological characteristics, self-interest and specific contingent needs and goals may, deliberately or not, prevail in jeopardizing the relationship and the animal’s well-being, leading to dysfunctional or even pathological interactions. Keeping wild species under unnatural conditions for leisure or personal gratification, exploiting farm animals through intensive farming causing them high levels of distress and pain, and abandoning domestic animals mostly for trivial reasons are only a few examples of poor and dysfunctional relationships that are topics of research and debate in the human–animal relationship literature [47,83,84,85].

The type of companionship sought by people may vary to a great degree, and companion animals may be acquired and kept mainly to fulfill different human needs and desires [8,86,87,88,89,90]. Some species, either unusual, rare or expensive, are kept as status symbols [87,91,92]; some breeds may serve as status objects and are appreciated just for their pedigree or appearance [93]. Companion animals, mainly dogs and cats, can serve as child substitutes or as toys [91,94,95].

Tuan [94] suggested that when pets are used as toys, they are treated capriciously to gain a sense of power and control that is also expressed through training them to obey to commands. Companion animals, especially dogs, can also serve as extensions of their owners’ self: they may extend their owners’ self not only symbolically by helping them to be something desired but also literally by providing them opportunities to do things that they could not otherwise do such as to engage in childlike games and playful activities or to extend their sphere of interpersonal relationships [88,96]. Keeping pets as an extension of the self implies that they are seen as expressions of the individual’s identity and as part of the person, and thus remaining without them is not conceivable; seeing animals as part of the self involves having an emotional attachment towards them and not just a functional one [97,98].

Finally, there are several examples of negative effects of the human–animal relationship, which include cruel acts and violence towards animals, with various characteristics and different degrees of severity (e.g., [66,81,99,100,101]).

Considering the human–animal relationship with all its nuances helps not only to gain a better understanding of the multidimensional and even contradictory nature of our interactions with other species but also to further explore the mechanisms involved in the “*hows*” and the “*whys*” of human behavior’’ [7,8,66,102].

## 4. Empathy, Attachment and Anthropomorphism: A Key Triad of the Human–Animal Relationship

Research provides compelling evidence that human’s relationships with animals cannot be easily characterized as positive–caring vs. negative–abusive but are variable and with several nuances. In the literature, three human psychological mechanisms are considered to play a key role in either positive or negative relationships with animals: empathy, attachment and anthropomorphism. These factors appear to be linked and influence each other, affecting people’s attitudes and beliefs towards animals and the way they are considered, treated and cared for [12,13,27,60,61].

Functional levels of empathy, attachment and anthropomorphism may promote people’s concern for animal welfare in general and foster healthy human–animal relationships [103,104,105,106]. Conversely, a dysfunctional attachment, the lack or suppression of empathy and no tendency to anthropomorphize are related to poor relationships with animals; higher acceptance of animal cruelty in children, adolescents and adults; a greater propensity to animal abuse; a reduced concern for animal welfare in general [107,108]. Dysfunctions in empathy, attachment and anthropomorphism may also lead to various types of animal abuse and neglect including animal hoarding [109].

### 4.1. Empathy


*“Sympathy beyond the confines of man, that is, humanity to the lower animals, seems to be one of the latest moral acquisitions (…). This virtue, one of the noblest which man is endowed, seems to arise incidentally from our sympathies becoming more tender and more widely diffused, until they are extended to all sentient beings. As soon as this virtue is honoured and practiced by some few men, it spreads through instruction and example to the young, and eventually becomes incorporated in public opinion.”*

*Charles Darwin “The descent of man, and selection in relation to sex”, 1871.*


As Darwin outlined, empathy is a key component of interpersonal relationships and a central aspect in our relationship with animals. The capacity to experience empathy is important in determining the level of concern and care that people have for other people, for companion and farm animals and for the conservation of wildlife and natural habitat [110,111,112,113]. For example, studies comparing individuals within animal protection and vegetarians, who share the purpose to avoid cruelty towards animals and to protect them, to general community samples reported that subjects from the first two samples had better attitudes towards the treatment of animals and enhanced empathic brain response towards them than others [32,114,115].

Empathy is considered a fundamental component of human emotional experience with an essential role in human social life and interactions [116,117]. It promotes social interactions, motivates prosocial behavior and caring for others (humans or nonhumans), inhibits aggression and is an affective/cognitive prerequisite for moral reasoning and behavior [118,119]. The capacity to empathize seems to be so relevant for a healthy coexistence that its absence or deficiencies are associated with socio-emotional problems and psychopathy [116,118,120].

Despite its many different definitions [121], empathy is considered a complex psychological construct comprising distinct but related components [116,122,123]. A basic component consists of affective empathy, which consists of resonance with others’ emotions and the generation of an immediate, appropriate emotional response [119,124].

Affective empathy entails experience sharing and the tendency to assume other individuals’ sensory, motor, visceral and affective states [125]. Affective resonance and emotional connectedness emerge early in human development and have a long evolutionary history of being shared across mammalian species [117,126]. Affective empathy has two different facets, namely, empathic concern and personal distress [127]. Empathic concern, is other-oriented, positively associated to emotion recognition and plays a central role in eliciting prosocial behavior, especially when associated with the understanding of others’ internal state [127,128,129]. Conversely, personal distress is self-oriented, can produce an aversive, self-focused reaction, with negative emotions, reduced emotion recognition and prosocial behavior [127,130].

The cognitive component of empathy involves recognizing and understanding others’ emotions, self/other awareness and perspective taking, i.e., the ability to understand what another individual is thinking or feeling [113,116].

The emotional and cognitive aspects of empathy cannot be easily separated, and their co-presence and normal development allow people to show compassion and sympathy and to engage in helping behaviors [116,120,121].

Although empathy can be automatically triggered, it is also modulated by top-down control processes and influenced by factors such as gender [120,131,132,133]—with women being generally more empathic towards people than men—motives, experiences, relationship with others, education [134,135,136] and contextual factors [125,131].

Culture is reported to attune the perceptual, cognitive and emotional processes involved in empathy to culturally determined ways of expressing emotions, pain and suffering, with cultural similarities and shared experiences modulating empathic responsiveness [137,138,139].

Human-directed empathy is amplified by perceived similarity (in appearance, personality and racial group) and familiarity (social closeness and previous positive experiences), and it is significantly reduced for those who are viewed as different, strangers or betrayers [123,126,140,141]. Perceived differences combined with labeling processes may promote the infra-humanization or dehumanization of others, i.e., a diminished attribution and consideration of their mental states [142,143].

Finally, people’s motives and goals affect perception, information processing and affective states including the willingness to empathize and shaping behavioral responses towards others [125,144,145]. In general, people are motivated to avoid highly distressing situations and to downregulate costly empathy, engaging in different strategies such as avoiding distressing situations that might trigger empathy, shifting their attention away from potentially affective stimuli or modifying their cognitive evaluations [146,147]. Devaluating others’ minds and emotions reduces empathy and justifies harming or killing others by minimizing their capacity to suffer or derogating them as deserving of this suffering [142].

In conclusion, the human ability to empathize and its variations depend on a complex interaction between different factors. Interestingly, many of these factors also appear to account for empathy towards animals and its variations [148], and it has been suggested that they could be associated [149].

Empathy towards animals seems to have originated in a similar way as that shown towards other humans [150,151]. It has been proposed that animal-directed empathy may generalize to human-directed empathy [152] and that the amount of empathy towards animals may indicate a more general capacity for empathy and related prosocial behavior [153].

Empathy towards animals could be a psychological “side effect” of adaptive empathy towards humans, triggered by animals’ signals, behaviors and physical characteristics that resemble those promoting empathy and caring towards humans, particularly infants [154,155]. Empathy probably evolved in the context of parental care, and its affective component covaries with the cute response elicited by the “baby schema” [56,126,154,156]. Infant-like animals trigger empathic responses and various studies have linked cuteness to increased empathy and compassion [157,158,159,160] and caretaking [161,162]. There is evidence, for example, that humans find one-day-old chicks, kittens and puppies cute and value nonhuman faces with infant features, such as those of puppies and kittens, as attractive as baby faces [17,56,77,163,164]. The existence of a biological mechanism deeply rooted in parenting that could account for empathy towards animals is also suggested by the large diffusion of pet-keeping and interspecific nurturant behavior, which are general human traits [42,77,154].

Human and animal-oriented empathy share several features including a gender effect, with women being more empathetic towards animals [13,27,33,34,113,114,136] and less likely to engage in animal cruelty than men [29,165,166], and a similarity and familiarity effect, which are good predictors of empathy for wildlife, farm animals and companion animals [55,140].

The similarity bias has been observed at different levels including phylogenetic closeness, physical appearance, behavior and infant-like features. The greater the similarity of a species with humans, the larger the empathic response, detected both through self-report and psychophysiological measures [60,140,167]. Westbury and Neumann [140] used film stimuli depicting humans, primates, quadruped mammals and birds in victimized circumstances and found higher subjective empathy ratings and physiological responses (i.e., skin conductance) as the stimuli became closer in phylogenetic relatedness to humans, with an effect of animal type (i.e., human, primate, companion mammal, utilitarian mammal and bird).

Familiarity with animals allows people to establish emotional bonds with them and facilitates the understanding that animals are living organisms, allowing to directly perceive the similarities between us and them, which are essential to empathize with them. For example, Morris et al. [168] reported an association between familiarity with animals in terms of ownership and beliefs about emotions (both primary and secondary) in animals and the animal mind in general.

Although most studies do not really differentiate between the affective and cognitive components of animal-oriented empathy, the ability to recognize animal emotions and to appreciate their communicative aims could be ascribed to the latter one. The capacity to correctly detect signs of pain and distress in animals is at the basis of animal welfare in companion and farm animals [169,170]. However, people’s capacity to recognize and understand animals’ emotional experiences and attempting to view situations from their perspective has variable levels of accuracy: when trying to “think like an animal”, humans tend to project human thoughts, feelings and attributes onto animals, often without sufficient experience or knowledge of the animal’s real biological and ethological needs, cognitions, emotions and behaviors [148].

Bradshaw and Paul [154] suggested that anthropomorphism is an expression of the cognitive component of interspecific empathy, and [13] reported significant correlations between anthropomorphic beliefs and empathy towards animals, suggesting that anthropomorphic interpretations could facilitate perspective taking and, consequently, the affective empathic reaction. However, anthropomorphism may also hinder people’s ability to “accurately” empathize with animals, leading to incorrect and flawed cognitive empathy with adverse effects for both animals and people [148].

Both human-directed and animal-directed empathy are modulated by contextual, social and cultural factors as well as by individual differences and experiences, which interact in a complex way with biological predispositions. For example, some human psychological traits, such as a need for power and hostility, are negatively related to empathy towards animals. The need for power leads to a utilitarian view of people and animals as a means for self-gratification rather than as living beings worthy of respect and concern, whereas hostility causes a temporary reduction in empathy, enhancing aggressions and reducing sensitivity to animal suffering and maltreatment [171,172]. There is also evidence that a lack of empathy is a characteristic of the psychological trait labeled as “callous and unemotional” and an association between “callousness” and animal abuse during childhood and adolescence has been reported [173,174].

Education and cultural background play an important role in fostering empathy towards animals. Paul and Podberscek [136] found that veterinary education affected students’ attitudes and empathy towards animals and that veterinary students in their later years rated the sentience of animals as lower than those in their earlier years. Similarly, a difference in empathy towards animals in Italian veterinary students was reported in [33], with first-year students scoring significantly higher than those at the end of their academic training. The decrease in empathy over time emerged in both male and female students, but females always had higher empathy scores than males. In addition, veterinary students at the end of their course reported a more instrumental attitude towards animals and a reduction in the perception of human–animal continuity, more evident in males than females.

The values, ideologies and social practices typical of a culture together with the social status attributed to animals affect the development of empathy and its behavioral expressions [175]. What is common to most cultures is the ambivalence towards nonhuman–animals, which are relegated to different cultural categories based upon species, for example, “food”, “companion”, “research tool” or “wildlife” [176]. Pallotta [176] noticed that young children are oblivious to the moral distinctions among different species of animals, which are learned during the socialization process; through normal socialization, children learn to place boundaries between themselves and all other animals and between different species of animals in terms of norms, emotions and moral treatment, and to canalize their empathy towards conspecifics and species with a higher social status.

Childhood socialization and cultural conditioning are generally mediated by parents; thus, attitudes and empathy towards animals are developed at first in the family setting through parental modeling: for example, children and adolescents often begin to abuse animals by reproducing the behavior of a parent who exerts a violent and coercive “discipline” on pets [152].

Finally, various studies indicate an association between violence towards animals and a lack or suppression of animal-directed empathy [28,152], with empathy representing a mediating factor in aggression towards both humans and animals [107]. Even though no mental disease has been specifically related to a lack of empathy towards animals, “hurting animals” is included among the diagnostic criteria of conduct disorder, and the last version of the DSM (2013) includes a psychological disorder, animal hoarding, which has been related to impairment of empathy and attachment towards animals [11,99,177,178].

### 4.2. Attachment


*“It is certain that associated animals [i.e., those living together in social groups] have a feeling of love for each other which is not felt by adult and non-social animals.”*

*Charles Darwin (1871). The Descent of Man, Vol. 1, p. 76*


Forming emotional bonds and attachments with significant others is a typical characteristic of human beings and represents a profound need with biological bases and evolutionary roots [179,180,181,182]. The need for bonding is shared by various nonhuman species, particularly social animals: mammals but also birds may form intense, long-lasting emotional bonds and attachments in the context of the parent–offspring relationship and in reproductive relationships between unrelated adults. In addition, there is evidence of friendship bonds between unrelated adult individuals of the same group; such bonds have been observed in nonhuman primates, dogs, horses, cows and even crows [18,19,183].

Emotional and attachment bonds go beyond the species boundaries, fostering significant, reciprocal interspecific relationships including the human–animal bond. Some studies show that people form strong affective bonds with their companion animals, reporting attachment to them and often viewing them as family members or even children [14,184,185]. Experimental evidence shows that dogs and cats form affectional bonds and even attachments with their human partners [24,186,187,188,189,190].

The human–animal bond implies emotional, psychological and physical interactions between people, animals and the environment, and according to [191], well-developed bonds are relationships between a human and an individual animal that are reciprocal, persistent and tend to promote well-being for both parties.

The concept of attachment, developed in the context of human interpersonal relationships, appears to fit well [191] the definition of the human–animal bond, capturing its different aspects. This concept was initially used by the psychologist J. Bowlby, within his ethological theory, to explain the nature of the bond that develops between human infants and their mother/caregiver [192,193,194]. Since then, attachment theory has been broadened to include other types of human relationships across the life span such as close friendships and romantic relationships [195]; more recently, it became a framework for investigating the human–animal bond [14,24,25,184].

In Bowlby’s perspective, attachment is a particular emotional bond that a person or animal establishes throughout life with another individual perceived as stronger or wiser [180,192]. Attachment bonds endure over time, are emotionally significant, are directed to a specific individual (attachment figure) and trigger proximity, contact seeking and distress reactions when unwanted or prolonged separations occur. Attached individuals seek security and comfort in the relationship with their partner, which serves both as a “secure base” from which to move off to navigate the world and as a “safe haven” to go back to in times of distress [180,181]. Attachment behaviors are species specific and organized into an attachment behavioral system that is activated to elicit appropriate caregiving responses [181,192,196].

In humans, attachment plays a key role throughout an individual’s lifespan, with either positive or negative effects on interpersonal relationships [197] and on interspecific ones [185,198]. According to attachment theory, early experiences with primary caregivers (parents or other “attachment figures”) affect an individuals’ future interpersonal relationships and stress regulation modalities through internal mental representations (or internal working models) of themselves, others and self–other relationships. In childhood, adolescence and adulthood, internal working models determine the degree to which individuals view themselves as lovable and deserving affection and others as trustworthy, reliable and affectively responsive [193,196,199]. Internal working models appear to mediate the link between attachment, empathy and prosocial behavior [196,199].

If caregivers are responsive and sensitive in distressing situations, the individual develops a secure attachment, which is associated with positive representations of self and others, the ability to manage distress, a sense of comfort with autonomy and the formation of relationships with others; attachment security promotes prosocial attitudes and empathy and has been associated to a better mental health both in adolescence and adulthood ([200]; see [201] for a meta-analysis).

Insecure attachment develops when individuals face insensitive or unresponsive caregivers [196,199]. An insecure anxious attachment is associated with negative models of self and a highly demanding interpersonal style, combined with fear of rejection and high levels of negative affect. Attachment anxiety is associated to the inhibition of empathy and the strengthening of personal distress, as anxiously attached individuals are too self-focused to provide help to another person [200]. Conversely, insecure avoidant attachment is associated with a negative image of others, defensive minimization of affect, interpersonal hostility and social withdrawal [202,203], and it appears related to the inhibition of both empathy and personal distress. When attachment figures are intimidating, unpredictable or frightening, the individual may develop a “disorganized attachment” [204] that leads to controlling behaviors that can be either hostile/punitive or solicitous/caregiving, affecting beliefs and behaviors towards people and animals and predisposing to personality disorders [205].

Insecure and disorganized attachments may favor compensatory defensive maneuvers that influence the caregiving system, have negative consequences on the individual’s well-being and mental health [206,207] and represent a risk factor for delinquency [208] and animal abuse and cruelty [106,209].

Studies on the human–animal bond based on the attachment theory indicate that the human–companion animal bond is an attachment-based relationship in terms of proximity, comfort seeking and separation distress but also as regards “safe haven” and “secure base” effects. This evidence extends the concept of “attachment figure” beyond the domain of human relationships to nonhuman social partners that, although unable to provide advice or focused support, can provide stability, tenderness, closeness, authenticity and absence of judgment [25,184,198].

Even though caregiving, protection and reassurance are usually provided by humans, the human–animal bond appears to be a more flexible attachment–caregiver relationship in which the human and the animal can play the role of “caregivers” or “cared for” according to the situation [14]. Companion animals may serve as “attachment figures” for people [182,184,185,198], and both dogs and cats form infant-like attachment bonds with humans, who are for them a source of protection and reassurance [24,186,188,189].

Indeed, several features of companion animals, such as their availability for direct physical contact, responsiveness to interactions and affection represent a strong basis for the attachment bond with the owner [182]. Zilcha-Mano et al. [185] showed that dogs or cats can serve the two main regulatory functions of an attachment figure: providing a “safe haven” and a secure base; however, a pet’s capacity to provide a safe haven and a secure base depends on individual differences in attachment orientations towards a pet, as found in interpersonal relationships.

The nature and structure of human–animal attachment appears to be similar to interpersonal attachment, with a significant association between security and insecurity in human–animal and interpersonal human relationships [105,198]. For example, an anxious human–animal attachment (i.e., pet attachment anxiety) was found to be associated with greater emotional distress and poorer mental health, ambivalence, pervasive worry for the integrity of the animal, doubt regarding owner’s worth for the animal [198] and a higher tendency for pathological grief [210]. Conversely, an avoidant human–animal attachment (or pet attachment avoidance) was associated with lower emotional distress, a relative indifference towards the animal’s integrity and needs [211], negative expectancies regarding the animal, a lower level of trust in the animal and a tendency to distance oneself from the animal [198]. Rusu et al. [212] reported the existence of significant positive correlations between pet attachment anxiety and interpersonal attachment anxiety and between pet attachment avoidance and interpersonal attachment avoidance in pet owners.

Notably, attachment difficulties with primary caregivers and attachment dysfunctions in adulthood are associated with cruelty and abuse towards animals [108,209], and several studies show that in animal hoarding, animal suffering and neglect may occur in conjunction with a strong distorted attachment to animals [11,213].

Attachment to animals is associated with empathy, attitudes and prosocial behavior towards them [13,39] and with anthropomorphism [65,212,214]. Rusu et al. [212] assessed the relationship between interpersonal and human–animal dimensions of attachment (i.e., anxiety and avoidance), empathy towards animals and anthropomorphism in owners of different types of pets (mainly dogs) and found that the level of anthropomorphism was positively associated with pet attachment anxiety and empathy towards animals and negatively associated with pet attachment avoidance. The authors suggested that animals may become emotional substitutes mainly for people with anxious attachment and worries about separation and abandonment who, in turn, attend more to the needs of their pets. Conversely, people scoring higher on avoidance in significant interpersonal relationships tended to be less attuned to the needs of their pets. Attachment anxiety and avoidance have been also reported to affect the decision to adopt a pet and the nature of the human–animal relationship such as the time spent with the pet and the perceived security of the bond with the pet [215].

Sociodemographic and cultural factors modulate attachment to animals both in terms of how attached people are and to which animals they become attached. Gender differences have been reported in various studies, with women showing higher attachment levels than men [12,216] and being more prone to providing verbal comfort and caregiving when their companion animals are distressed after a separation [217]. Cultural norms and beliefs towards nonhuman animals and anthropomorphism also appear to play a role in affecting attachment to and caring for animals [12,218,219].

### 4.3. Anthropomorphism


*“Believe me, I am not mistakenly assigning human properties to animals; on the contrary, I am showing you what an enormous inheritance remains in man to this day.”*

*K. Lorenz, King Solomon’s Ring, 1952 p. 152.*


It has been suggested that anthropomorphism is a key aspect in the formation of the human–animal bond and the practice of pet keeping, since it allows people to identify and address the needs and the psychological states of animals in a context of reciprocal beneficial interaction [149,214]. Even today anthropomorphism may have a positive role in fostering human–animal relationships and in promoting animal welfare, due to its power to affect the way in which people perceive, interact with, and respond to animals [104,220,221,222].

In the psychological literature, anthropomorphism has been defined as the human tendency to see human characteristics or mental states in nonhuman agents, either natural entities, objects or nonhuman animals, attributing them human intentions, motivations, goals or emotions [143,223]. Anthropomorphizing animals entails attributing them behaviors, personalities, mental abilities, emotions and intentions that are human like. Thus, it can be viewed as an anthropocentric bias in which humans use themselves as a benchmark for interpreting animal behavior [224,225]. Anthropomorphizing implies making more or less accurate inferences regarding others’ characteristics (e.g., affirming that a dog is feeling guilty) based on one’s own egocentric experience or on knowledge regarding humans in general [65,143,224]. Most animal scientists are highly concerned about the risks of anthropomorphizing, yet an anthropomorphic approach is invariably applied by people, and it has been proposed that it could be applied in ways that are useful to scientific inquiry and to the animals themselves [226,227].

It has been suggested that anthropomorphism evolved in humans due to the fact of its adaptive function to use self-knowledge to explain and anticipate other humans’ behavior [228,229]. According to [228], the propensity to attribute characteristics of our species to other animals is a feature of modern humans that emerged through natural selection approximately 40,000 years ago to favor our ancestors in hunting, making it more successful. In addition to enabling humans to predict and anticipate the behavior of prey animals, anthropomorphizing also encouraged the development of empathy for hunted animals and their offspring [103,154], which were then adopted and cared for, laying the foundations for the domestication of some species and the emergence of affectional relationships with animals.

Anthropomorphism is associated to concern over another nonhuman agent’s well-being and a greater likelihood of treating that agent as human [143], and there is a link between anthropomorphism, empathy and attachment towards nonhuman agents, particularly animals [13,103,212]. Thus, attributing human-like mental characteristics (i.e., emotions, cognitions and sentience) and features (e.g., human-like “face” or movements) to animals may strongly affect the way we consider and treat them and the moral concern we have for them [60,64,223].

Although widespread worldwide, the tendency to anthropomorphize animals is neither invariant nor stable: children are more prone to anthropomorphize than adults, some people anthropomorphize more than other people, some situations promote anthropomorphic beliefs more than others, and anthropomorphic descriptions of animals are more common in some cultures than others [65,143].

It has been outlined [65,143] that human anthropomorphism includes cognitive and motivational components and is modulated by individual factors (e.g., need for control and chronic loneliness), situational variables (e.g., perceived similarity and social disconnection), developmental factors (e.g., attachment and acquisition of alternative knowledges) and cultural aspects (e.g., norms, ideologies, individualism or collectivism).

Anthropomorphic thinking provides a sense of social contact and connection and satisfies the human need to deal with uncertainty and feel efficacious [65,143]. Humans are highly motivated to maintain social connection with others, and there is a strong association between morbidity–mortality and social connection [230]. People who feel lonely or chronically lack social connection with other humans may try to compensate by creating a sense of human connection with nonhuman agents, anthropomorphizing them; animals, particularly companion animals, such as dogs and cats, are easily anthropomorphized, since they show complex behaviors and may engage in active relationships and communication with humans [71,214]. Individuals who report feeling lonelier provide higher evaluations of the supportive anthropomorphic traits of their pets (e.g., thoughtful and sympathetic) than those who feel more socially connected [65], and the likelihood of attributing human-like mental states or traits to pets appears to be greater if individuals, in an experimental setting, are induced to experience a state of loneliness or social disconnection than if they are not [231].

Anthropomorphism also provides a practical way to interpret others’ behavior, particularly when alternative knowledge (e.g., science or culture) is lacking. Epley et al. [65] asked adult participants to look at a short video in which two dogs interacted with each other and one dog appeared less predictable than the other in behavior; then, participants were asked to rate the extent to which each dog was aware of its emotions, had a conscious will, had a “personality” and their similarity to humans. Participants with high scores in desire for control anthropomorphized the unpredictable dog more than those with low scores.

There is evidence of a relationship between anthropomorphism, attachment and attachment styles: people with insecure–anxious attachment styles with close others may compensate by seeking more secure or stable relationships from nonhumans agents, and companion animals provide stable, affectionate and nonjudgmental relationships [143,232,233,234,235].

Other variables that affect anthropomorphism are perceived similarity and phylogenetic relatedness to humans: people attribute mental states to animals that most resemble them physically or behaviorally or are perceived as more related to them [143,236,237,238,239]. Basic mental states and abilities are attributed more easily to a wide range of animals than complex mental states or higher cognitive abilities [239,240]. Similarly, primary emotions (i.e., fear, joy, surprise, sadness, anger and disgust) are attributed more frequently to a wider range of animals than secondary emotions (e.g., embarrassment, guilt, empathy, pride and jealousy) [224]. However, it has also been outlined that people perceive human characteristics in animals during interactions with them and within specific interactional settings [241,242]. This suggests that anthropomorphism would not depend just on the characteristics of a given animal but also on the kind of interaction and relationship between the person and the animal [242].

Arluke [243] provides several examples from biomedical laboratories, showing that an animal, for example, a rat or a dog, may be considered either an object or a pet depending on the kinds of actions in which humans involve them. The author highlights that animals may be strategically deprived of their individuality and expressive capacities using de-anthropomorphizing strategies aimed to objectify them (e.g., cages, codes and avoiding giving them a name). De-individualizing animals, treating them as a collective entity and labeling them with a code not only facilitates the redefinition of the animal’s nature but also materially prevents laboratory workers from seeing them as individuals. Similar strategies aimed at keeping animals in the “right perceptual frames” are adopted in intense farming to prevent developing familiarity, relationships and even attachment to them.

Experience and culture affect anthropomorphism by providing different norms and ideologies regarding how people relate to others, the natural world and animals, by influencing the general level of experience with certain animals and by the acquisition of nonanthropomorphic knowledge [12,143,218,219]. In less developed and rural populations, children show relatively little anthropomorphism when reasoning about local nonhuman animals [244]; individuals from more industrialized cultures think about nonhuman animals mainly based on their anthropocentric knowledge, whereas individuals from less industrialized cultures use their knowledge about the animal’s world. Finally, in individualistic cultures, an individuals’ egocentric perspective is used more readily than in collectivistic cultures [245].

Anthropomorphism has either positive or negative effects on human–animal relationships and animal well-being [69,103,104,221]. The tendency to anthropomorphize can promote positive relationships with animals, favoring attachment and empathy towards them [103,104,246]. However, anthropomorphism may also lead people to an anthropocentric and inaccurate understanding of animal’s mental abilities and to ascribe them cognitive and emotional capacities, intentions and needs they do not have, causing misunderstandings, unrealistic expectancies or disregard for their real needs and characteristics [69,224].

Various anthropomorphic practices result to be detrimental for companion animals’ well-being and health [69] and there is evidence that animal directed anthropomorphic behaviors and practices may be driven by temporary fashions or by different self-centered motivations, as the need for control, loneliness, satisfaction of one’s social needs and emotional attachment [8,12]. For example, the selection of specific physical and behavioral traits that favor the attribution of human mental states to nonhumans may transform companion animals (especially dogs and cats) in appealing but handicapped and unhealthy individuals [103]. Brachycephalic breeds are just an example of this dark side of anthropomorphism in which the exasperate selection for “cute” and infant-like body traits causes severe health problems and a poor level of well-being. Despite the well-documented unhealthy conditions of brachycephalic breeds, their owners are less influenced by breed-related health problems and reduced longevity compared with nonbrachycephalic dog owners in the decision to adopt a dog [93,247].

People view and treat nonhuman animals in line with classifications (e.g., companion animals, profit-making animals and wild animals) based upon ascribed similarity/difference with humans [248,249]; this entails different levels of anthropomorphizing. There is evidence that when people stop attributing human-like characteristics to other humans, they dehumanize them, treating them as nonhuman animals or even objects, and harming or killing them becomes easily accepted [142]. Similarly, anthropomorphic attributions affect the moral status given to animals and human moral concern towards them. Those that are believed to be more human-like are typically afforded greater moral consideration and better treatment than the others [63,64,240].

Although reasons for cruelty and abuse towards animals are wide ranging and complex [250,251] and with no specific reference to anthropomorphism, they have been traced back to the need for control over animals, prejudices against a particular species or breed and to simple dislike for an animal [252]. Some studies also found a significant positive association between anthropomorphism, hoarding behaviors and emotional attachment to possessions [234,253,254], and although the relationship between animal hoarding and anthropomorphism has scarcely been investigated, there is some evidence (e.g., [235]) that animal hoarders tend to anthropomorphize animals to a greater extent compared to non-hoarder animal owners.

## 5. Animal Hoarding: A Pathological Human–Animal Bond

Animal hoarding is a highly dysfunctional and pathological form of human–animal relationship, which has been defined by Patronek [11] as “the third dimension of animal abuse”, since it entails substantial and protracted animal maltreatment and suffering. Due to the fact of its characteristics, it cannot be easily incorporated into the two well-recognized categories of animal cruelty: deliberate animal abuse and neglect [11]. Indeed, in animal hoarding, animals are exposed to considerable physical and psychological suffering, but often there is a strong human–animal bond, with considerable impairment also of the hoarder’s welfare, who may lack insight regarding the real situation.

Since early reports [177,255], it has clearly emerged that the association between severe animal suffering and a strong attachment to animals was in contradiction with the existing evidence on the human–animal bond [11,213,256]. The evidence that has emerged so far invites scholars to an in-depth reflection on the boundaries between the “normal” and pathological aspects of the human–animal relationship and the mechanisms involved [8,99]. For a long time, hoarding animals was considered a “lifestyle” typical of bizarre and strange animal lovers, mainly women, but now it is considered a form of animal abuse, and in recent years, it has been recognized as a mental disorder acknowledged in the DSM-5 as a form of hoarding disorder (i.e., animal hoarding disorder (AHD) [257], although with its own characteristics and peculiarities.

Hoarding disorder (HD), or compulsive hoarding, is characterized at the behavioral level by problematic behaviors of accumulation, a persistent and considerable difficulty in discarding ordinary items, squalor, personal neglect and poor insight into such disruptive behavior. Hoarding behavior prevents the ordinary use of living spaces in the home, causing significant distress and impairing everyday functioning [257,258,259]. HD has been associated to various factors such as interpersonal conflicts, health issues [260], anxiety-based disorders, depression, family and social disabilities, progressive functional deficit [261], social isolation and difficulties in bonding with other people [262]. In addition, an association with cognitive deficits in attention, memory and executive functions (e.g., planning, decision-making and inhibitory control) has been reported [263,264,265]. Finally, compulsive hoarding is characterized by a problematic emotional attachment to possessions [258,266,267]. According to a psychological cognitive model, this emotional attachment has three specific aspects: possessions provide comfort and security, possessions have human-like qualities and possessions represent an extension of self-concept [266,268].

Animal hoarding (AHD), in its great complexity, appears to have common characteristics with object hoarding including possible underlying cognitive impairments [265], anxiety, depression, social isolation and relational difficulties [269,270]. However, there is a growing consensus that animal hoarding differs in several respects from object hoarding, the most striking being probably that animal hoarders accumulate living, sentient beings that, differently from objects, need interaction and attention and require continuous nurturing and care. Animal hoarding appears to entail distortions and dysfunctions in attachment, empathy and anthropomorphism and shows how characteristics and motivations that, in general, foster and maintain long-lasting positive relationships with animals may jeopardize the human–animal relationship, causing suffering for both animals and humans [99,178,235].

Animal hoarding occurs when an individual persistently accumulates an unusually large number of animals and fails to provide them with the minimum standards of nutrition, hygiene and veterinary care, exposing them to psychological and physical suffering due to the fact of a seriously distorted relationship with them [99,271].

Hoarders often fail to recognize and act on the deteriorated conditions of the animals (e.g., disease, starvation and death), the severe overcrowding, the lack of hygiene of the home environment, and they are often unaware of the negative effects that the hoarding of animals has their own well-being and on that of others living close to them [177,272,273,274]. Squalid living conditions are common including extreme soiling, parasites, litter, precarious waste, accumulation of feces and urine, nonfunctioning bathrooms and other living spaces, and even dead animals left where they died or are stored [11,275]. Thus, animal hoarding is a multifaceted problem that includes animal maltreatment and abuse, problems related to the health and mental health of hoarders, safety and social and occupational functioning [177,255].

It is worth noting that what defines the disorder is not just the number of animals but the inability of the owner to provide the minimum necessary care for them, to recognize their suffering and to provide them with appropriate sanitary conditions [177,235,276,277,278]. Animal hoarders’ difficulty in relinquishing animals to people who can more adequately care for them is another key point: despite highly negative conditions they form excessive attachments to their animals, consider them like children [279] and feel the urge to save and care for them, even if this results in significant impairment. They also exhibit intense distress when the animals are removed from their care [235,266,275].

Despite its complexity and the highly negative impact on both animals and people, animal hoarding has long been underestimated both within and outside the academic community; however, recently, interest has grown in different research areas such as mental health, psychiatric disorders, human–animal relationships and veterinary practice [99,272,280,281,282].

Animal hoarding is not limited to a specific culture or country, and in addition to the US, it has been reported in various countries, including the UK [283], Canada [278,284], Australia [274,285], Serbia [286], Spain [273], Italy [281], and Brazil [276,287], and can represent a significant problem [265,272,288]. For example, [288] evaluated the situation of animal hoarding in Germany, reporting 120 animal hoarding cases between 2012 and 2015 for a total of 9174 hoarded animals, mainly cats, dogs and small mammals. The highest number of accumulated animals was documented in [281] in Italy, with a total of 450 hoarded animals.

Some reviews of the literature provide a portrait of the main characteristics of animal hoarders and the factors that may be involved in the emergence of the disorder [99,282,289]. Overall, the studies outline that animal hoarding is a chronic disorder that progressively deteriorates [273,274]. As regards an animal hoarder’s characteristics, the literature shows that the disorder is more frequent among women, who constitute, on average, between 70% and 83% of animal hoarders [99,282,289]. This rate has been confirmed in most of the reported hoarding cases (e.g., [265,273,285,290]). Although the reasons for this gender effect have not yet been ascertained, it could be related to the greater predisposition of women to empathize with animals and to be attracted by the infantile characteristics typical of most pets, especially dogs and cats, which are in fact the species most involved in hoarding situations. Gender differences in attitudes, empathy and attachment and concern towards animals have also been well documented in the literature on the “bright side” of the human–animal bond [13,29].

Individuals showing animal hoarding are reported to be socially isolated, in their fifties or sixties, on average, when identified [265,273,274,276,285] and in approximately 70% of the cases are single, divorced or widowed [265,276]. Studies also show that hoarders are often unemployed or retired but may also be breeders who initially bred animals for economic reasons [274,276,285,290,291]. However, the phenomenon emerges in all demographic socioeconomic conditions [99,282,289]; even individuals who are well integrated into society may be affected (e.g., health care workers, public employees, lawyers and veterinarians), and not uncommonly hoarders live with people who depend on them, including children, people with disabilities or elderly people, who may find themselves sharing the same living conditions as the animals being accumulated.

Three main types of animal hoarders have been described in the literature, who show different characteristics and different levels of severity [177,292]. These characteristics regard the presence of medical/psychological problems, involvement in society, awareness of the problem, attitude towards authority and risks for animals but also attachment to animals, empathy towards them and the tendency to anthropomorphize them [99,235].

The “overwhelmed caregivers” [177,292] are usually lonely people with a strong attachment to and empathy for animals, who initially provide proper care to them but eventually become overwhelmed due to the fact of difficulties such as illness, economic problems or a bereavement. Their self-esteem is linked to the role of caregiver, and they often adopt animals in a passive way (people give them animals knowing they love them). These hoarders may have mood disorders but show a certain level of awareness of their problems, respect authority and are cooperative.

For “rescue hoarders” [177,292], saving animals is a mission and they are convinced they are the only ones who can provide adequate care to them. These hoarders have a need to acquire animals actively, adopting them from shelters or through flyers or social networks. After rescuing animals, these hoarders prevent their adoption, are afraid they could die and are opposed to euthanasia even when seeing them suffering and healing them is impossible. Thus, the number of animals gradually overwhelms their capacity to provide the minimal care. They are not necessarily isolated but may have a network of helpers that provide them animals to care for. Indeed, these hoarders may be found among people working in rescue shelters or in veterinary clinics, who believe they can save all animals by taking them home [291]. These types of hoarders avoids authorities and/or impedes their access, making the solution to the problem difficult.

The last type reported in the literature is the “exploiter hoarders” [177,292], who acquires animals to serve their own needs (e.g., absolute control and demonstration of expertise). They lack empathy for people/animals, are indifferent to the harm caused to animals or people and are frequently manipulative and devise strategies to avoid controls. They tend to deny the problem and reject concerns from any authorities over the animals’ care. However, it has been outlined that exploitative hoarding is often associated with sociopathic traits or personality disorders, either narcissistic or antisocial, that resemble much more those of people who engage in criminal behavior and animal abuse and cruelty.

Overall, the literature indicates that a hoarder’s story begins with relatively few animals, and that gradually, the situation deteriorates through the continual acquisition of animals (actively or passively) despite a lack of money and time for them, avoidance of sterilization, insufficient veterinary care and the struggle to keep the house sufficiently clean. The incipient hoarder and the breeder–hoarder are considered intermediate stages that may evolve over time into full-blown hoarding [292,293]. Different from an incipient hoarder, a breeder initially keeps animals for shows or to sell but over time finds it increasingly difficult to care for them properly. They do not always keep the animals in their own home; thus, their living conditions may not always be as neglected as those of the animals. However, this is not the rule, since many hoarders set-up home breeding. They usually have only a moderate awareness of the state of the animals and their ability to care for them; thus, they continue to raise them.

### Empathy, Attachment and Anthropomorphism in Animal Hoarding

In general, animal hoarders show a highly dysfunctional relationship with animals, characterized by distorted overattachment to them, empathy disfunction and a high level of anthropomorphism [99,235,289]. Their behavior, especially that of incipient hoarders, overwhelmed caregivers and rescue hoarders, is most often rooted in a real desire to take care of animals or save them and is characterized by a strong emotional attachment to animals and an extreme distress, often genuine despair, at the idea of being separated from them. Patronek and Nathanson [294] reported that when attempting to account for their behavior, these hoarders often mention their love for animals. However, the situation degenerates from helping into a form of abuse such that, even though the intent is not to harm animals, severe and prolonged suffering is caused to them [275].

In most hoarding situations, the emotional attachment to animals, which is a basic aspect of the human–animal relationship, is distorted, formed with many individuals and immediately triggered to the point that any animal encountered can be easily seen as one’s own and the individual feels obliged to take care of it [295].

It could be hypothesized that empathy (in particular affective empathy) could be the initial mechanism that makes these people sensitive to the needs and suffering of animals, motivating them to take care of them; however, when facing the evidence of their inability to care for animals properly, a denial process would take place to protect their sense of identity and self-esteem, both strongly rooted in the link with animals, in the belief of a special connection with them and in the presumed ability to take care of them. Furthermore, empathy could decrease as a result of exposure to suffering due to the fact of habituation and/or to avoid an excessive level of personal distress or even to dealing with many individuals and a reduction in personal contact [134,135]. Conversely, “exploiter hoarders” generally lack empathy for animals and people, being more similar to people with an antisocial personality and who exhibit such behavior and are prone to animal abuse, in whom empathy towards animals and people is compromised in some way [118,296].

Hoarders also tend to anthropomorphize animals to a greater extent compared to ordinary animal owners. Steketee et al. [235] observed that 81% of animal hoarders (compared to 27% of owners) tended to ascribe to them the same characteristics and intelligence of humans and to view them as their “children”. In fact, they often report a strong attachment to their animals and consider them to be like children [279]. This often results in a distorted sense of responsibility and a strong need for control over the animals, whereby the hoarders feel that they must acquire animals and should not separate from them to ensure that nothing bad happens to them.

In “rescuer hoarders”, these aspects would be strongly linked to the problem of death: they appear to consider the deterioration of their living conditions as a necessary sacrifice to help creatures in need, who might otherwise die [295], and some of them explicitly state that they would like to create shelters that do not include euthanasia [297]. Thus, there is a real urge to rescue animals, which is experienced as a duty, leads to a strong sense of guilt if disregarded and is linked to the constant concern that something terrible could occur to the animals if they were not helped (e.g., being hit by a car or ending up in a vivisection laboratory) [295]. Since animals, as all living beings do, die and separation from them is inevitable, anxiety relative to control and responsibility is further enhanced in hoarders [294]. These individuals may have intense emotional reactions of anger or distress triggered by thoughts of loss or separation [264] and often are unable to separate themselves from the bodies of dead animals, keeping them inside the house [177,255,275]. Based on these extreme reactions of separation-related distress and anger, Reference [298] hypothesized that the propensity of animal hoarders to ignore the problems arising from acquiring an increasing number of animals and to convince themselves that the animals are well might be a way to avoid the unpleasant feelings that would result from giving animals up for adoption or from acknowledging the severely poor conditions of their animals.

Emotional attachment, concern for animals and empathy are less explanatory in the case of “exploiter hoarders”, whose motivations for hoarding could be linked to a need to dominate and control or to financial interest (especially in the case of anti-social personalities) or to the desire to establish relationships that confirm their value, in which other individuals (in this case animals) have the role of self-objects, serving to ensure attentions and adoration (in narcissists; [292]). Similar motivations, however, can be observed in a far more nuanced form also in so called “normal” non abusive human–animal relationships [8,88,94,299].

For animal hoarders, animals also have an instrumental role, which is functional to the preservation of their sense of identity and self-esteem, associated with the role of caregiver and is constantly reinforced by the perception of having positive relationships with living beings that are sentient and totally dependent [294,295]. Self-esteem would derive partly from the sense of self-efficacy—through the control over the animals (especially in the exploiter) and in part by seeing recognized their ability as caregivers through the affection received by the animals, [294,300]. However, the primary source of self-esteem is the interaction within the human–animal relationship: animals are highly focused on the person and do not judge or criticize, and they cannot object to misinterpretations of their feelings and needs [294].

The central role of animals in fostering and maintaining an individual’s self-esteem and sense of identity has also been documented in studies on normal human–animal relationships and are one of the psychologically positive effects of the human–animal bond [71,301,302].

Like non-hoarders, incipient hoarders, overwhelmed caregivers and rescue hoarders derive a sense of security and comfort due to the fact of animals’ capacity to provide emotional support and unconditional love in a relationship that is perceived as less dangerous than those with other people [275,291]. Animal hoarders are reported to have difficulties in establishing affective bonds with others, tend to maintain social isolation [235,270] and prefer contact with animals [275,291]. They could consider their relationships with animals as safer and more rewarding than interactions with humans [275,294]. Cats and dogs are the most accumulated species [278,289] but also the two most common species of companion animals; they have a long history of domestication and close association with humans, live close to them and are widely considered as important social partners by their owners in many countries. However, hoarded animals may include a variety of animals, such as miniature ponies, deer, ferrets, pigs, various species of birds, and even spitting llamas, and multiple species may be present in any isolated hoarding case [278].

Steketee et al. [235] interviewed individuals who fit the criteria for animal hoarding and individuals owning many animals but not meeting the hoarding criteria, reporting that both hoarders and non-hoarders had stressful life events in childhood and adulthood, strong feelings about animals such as the desire to rescue, take care of and be close to them. Animal hoarders, however, had more dysfunctional interpersonal relationships and mental health concerns and anthropomorphized animals more often; however, the most significant difference between the two samples was the presence of a chaotic domestic environment during childhood and childhood problems with caregivers (e.g., unstable, neglectful, abusive, absent, and/or inconsistent parents). In another study [102], it was analyzed whether people owning 20 or more cats shared the commonly reported psychological and demographic profile of animal hoarders compared to owners of 1–2 cats drawn from the same population. They found that people who owned many cats were more similar to clinical animal hoarders in age and pet attachment levels than the typical cat owners, but they differed in functioning, veterinary care and home organization.

The quality of caregiving and attachment during infancy seem to play an important role in the emergence of animal hoarding, and animal hoarders often report that during childhood they relied on companion animals, suggesting that in difficult developmental situations companion animals may function as alternative attachment figures providing intimacy and security without fear of rejection [303]. Nathanson and Patronek [109] hypothesized that when parenting was neglectful, inconsistent or abusive, animals became crucial in animal hoarders’ childhood to maintain and promote empathy, comfort, calm, acceptance and self-esteem, as previously suggested by Brown [97,98,304] in the theory of animals as self-object. Secure attachment in early childhood is considered essential for normal emotional development, emotion regulation, empathy and good interpersonal relationships [205,303] and attachment to pets plays an important role in normal social, emotional and cognitive development, promoting mental health, well-being and quality of life [14,21].

Due to the occurrence of developmental and life events, animal hoarders may develop in adulthood a compensatory over-reliance and an overattachment to animals, with animals becoming a dysfunctional solution to cope with their need for relationships and intimacy without fear of rejection and abandonment [294]. An abusive, traumatic or dysfunctional childhood is correlated with a disorganized attachment style, which can result in compulsive caregiving. This caregiving style, differently from the sensitive caregiving style (i.e., the ability to be responsive and attuned with another’s individual support-seeking behavior), is characterized by the tendency to provide care obsessively and intrusively, irrespective of whether the care is wanted or needed [193,305,306]. In adulthood the compulsive caregiving of animals can become the primary way to maintain or building a sense of self. This kind of controlling behavior often characterizes the caregiving style of animal hoarders, together with other forms of control typical of animal hoarders including refusal to adopt, rejection of help and expert opinions regarding proper animal care and sometimes the saving of dead bodies [294].

Probably, one of the most perplexing aspects of animal hoarding is that animal hoarders declare to love animals and want to care for them but, in fact, their animals are terribly neglected and suffering. This lack of insight is typical of animal hoarders, and it has been suggested that being unaware of the degradation in their personal lives and those of their animals could be suggestive of dissociation [294].

Dissociation represents a self-protective strategy to avoid negative feelings associated with distress or trauma [307] and can make it difficult to understand and respond to others’ feelings as well as easier to view them as less than human [303]. In the case of animal hoarders, dissociation would represent a strategy to preserve the integrity of the self, the self-image and the mission of caregivers, notwithstanding the extremely precarious conditions of their animals.

It has been argued that dissociation may be best understood as a continuum from more common, normal manifestations to less common and more pathological symptoms [308]. Indeed, in [309] it was reported that in a sample of college students, dissociation was positively associated with attachment to companion animals, and this result was then replicated in another student sample [310].

## 6. Conclusions and Afterthoughts

The literature on human–animal relationships highlights the complex and highly multifaceted nature of this relationship showing that it may have positive, nonfunctional and clearly detrimental effects on animals or, in some cases, both animals and people. Empathy, attachment and anthropomorphism are considered to play an important role in either positive or negative relationships with animals and influence each other in determining the type and quality of the relationship between humans and animals.

Animal hoarding is a highly dysfunctional and pathological form of human–animal relationship, provides a striking example of the great variability in the way people may relate with animals and shows that psychological processes and motivations that promote and maintain positive and healthy relationships with animals may also have highly negative effects, causing suffering to both animals and humans. Patronek [11] defined animal hoarding as “the third dimension of animal abuse”, underlining that it entails substantial and protracted animal maltreatment and suffering but cannot be easily labeled as deliberate animal abuse and neglect due to the presence of a strong, albeit detrimental, attachment and human–animal bond.

Interestingly, as do most owners of companion animals, animal hoarders appear to have a strong emotional attachment to animals that represent a source of comfort and security and also provide a sense of social connection and self-efficacy. As in non-hoarders, in animal hoarders empathy (especially affective empathy) is a mechanism that triggers sensitivity to the needs and suffering of animals, motivating them to take care of them.

Furthermore, animal hoarders, similar to ordinary animal owners, attribute animals with human-like qualities [235], deriving comfort and well-being from their ability to provide emotional support and unconditional love and may view them as an extension of their self-concept [294].

It has been proposed that viewing animals as extensions of themselves, rather than separate beings, would make hoarders unable to empathize with them or to understand that they have needs of their own [294,304]. However, although to a different degree, even in nonpathological relationships companion animals, especially dogs, can serve as extensions of the owners’ self [88,96], helping them become something desired or giving them opportunities to do things that they could not otherwise do [96].

Similar to common pet owners, animal hoarders report that during childhood they relied on companion animals, suggesting that companion animals represented for them attachment figures, providing predictable intimacy and security without fear of rejection, maltreatment or abandonment. Then, in adulthood, hoarders may develop an abnormal over-reliance on companion animals for attachment and support, and animals become a dysfunctional strategy to cope with their need for relationships and intimacy.

People who hoard animals, depending on their prior life history, type of attachment developed and other conditions that emerge both at the intrapsychic and environmental level, may develop different forms of animal hoarding with different peculiarities. Indeed, the evidence on animal hoarding indicates that this syndrome is very complex and with great heterogeneity [100,292].

Yet, pet ownership is also complex, multifaceted and varies along a continuum. There are owners forming functional, healthy relationships with one or relatively few animals—where the needs of both owner and animals are met—and owners forming dysfunctional relationships with their pet, where the needs of the pet are disregarded and only the owner’s convenience and needs are pursued [8]. There are also owners keeping larger numbers of animals that are superficially comparable to clinical animal hoarding cases, but animal and human welfare are not really compromised [102].

Future studies comparing different types of companion animal owners with animal hoarders—controlling for other variables such as age, gender and the types of animals involved—would be useful to gain further insight into the complex, multifaceted nature of the human–animal bond and its distortions in animal hoarders.

As attachment, empathy and anthropomorphism appear to be basic ingredients of the human–animal relationships, it would be interesting to further investigate the differences in attachment, empathy and anthropomorphic thinking between normal pet owners with different types and number of pets and animal hoarders, as pioneered in a few studies so far [102,235].

Hoarders’ instantaneous attachment to animals and its negative effects could be related to hoarders’ higher sensitivity to the infantile features of pets. In addition, their attachment to a large number of animals and their willingness to rescue them could depend on a greater empathy towards them, associated with dysfunctions of some regulatory mechanisms, entailing similarity and familiarity, or to errors in the ability to process animals’ emotions, with stimuli-like facial expressions and vocalizations being interpreted in a systematically negative way (leading the hoarder to perceive animals as always in need of help), as observed in individuals who, like animal hoarders, have experienced childhood trauma or have personality disorders [311].

Another aspect that could be further investigated in animal hoarding is the role of anthropomorphism. The potential relationship between anthropomorphism and hoarding is intriguing and could help to further explain the strong and quick attachment animal hoarders form with animals and their distorted empathy.

Anthropomorphism is a pervasive human characteristic. It is found at different levels in normal human–animal relationships and has been shown to be important even in the hoarding of objects [253].

Finally, it would be interesting to gain more knowledge on the characteristics of the different forms of animal hoarding, in particular further investigating to what extent the “exploiter” hoarder, who lacks empathy, attachment and concern for animals, overlaps with other forms of psychopathology such as the antisocial personality disorder. This type of hoarder, who exploits animals, appears to also include individuals who derive profits from an intentional improper management of breeding farms and animal shelters, and considering them as just hoarders could erroneously turn a form of criminality into a psychological disorder such as animal hoarding.

## Data Availability

Not applicable.

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
