# Peer review of "The Complexity of the Human–Animal Bond: Empathy, Attachment and Anthropomorphism in Human–Animal Relationships and Animal Hoarding"

_animals, 2022, doi:10.3390/ani12202835_

Round 1
Reviewer 1 Report
This paper is a review of literature on human-domesticated animal relationships framed by consideration of the roles of empathy, attachment, and anthropomorphism. The paper asserts that the "dark" side of these relationships has not been adequately addressed. This is arguable as, although the benefits of HARs have been more thoroughly researched, the literature on the "link" (the dark side) is considerable. In any case the paper does not get to the dark side until page 14.
This is a nice review of the literature but, in my view, does not make a significant contribution to the literature. There are other reviews available. The frame using the three psychological processes mentioned is not particularly informative; nor is its application to hoarding.
I recommend major revision, greatly reducing the review of the benefits of HAR literature and, either reducing the implied claim that the three processes constructively reframe the literature on this subject or more convincingly showing that it does.
Author Response
Responses to Reviewer 1
Rev1. This paper is a review of literature on human-domesticated animal relationships framed by consideration of the roles of empathy, attachment, and anthropomorphism.
Response. Our aim was not to make a review of the literature on human-domesticated animal relationships, but to outline the nuanced nature of the human-animal relationship, with a focus on empathy, attachment and anthropomorphism as key factors in determining it. Thus, we did not review all the available literature on the issue. Since it is possible that this aim was not introduced clearly enough, we have now clarified this in the title, the abstract and the introduction (Lines 17-20; 33-37; 56-63).
Rev1. The paper asserts that the "dark" side of these relationships has not been adequately addressed. This is arguable as, although the benefits of HARs have been more thoroughly researched, the literature on the "link" (the dark side) is considerable.
Response. Probably there was a misunderstanding and possibly we were insufficiently clear since we did not intend to assert that the dark side of the relationship has not been adequately addressed so far and in fact there was no such statement in the paper. We agree that there is a large literature on the the "link" (the dark side) and our aim was not to review this literature. As it seems that the use we made of the term “dark” (and “bright”) has been a source of misunderstanding, we have decided to remove the term “dark” from the title, the sections’ titles, the abstract and throughout the text. Thus, in the revised version we have replaced the term “dark” with negative, dysfunctional or leading to animal violence, abuse or neglect as we do not intend to make direct reference to the "link.
In addition, two sentences have been modified in the text (Lines 159-163) to make it clear that both positive and negative aspects of the human-animal relationship have been thoroughly researched.
Rev1. In any case the paper does not get to the dark side until page 14.
Response. As specified above the term “dark” has been now removed to avoid misunderstandings. In any case we would like to underline that in the submitted version of the paper the issue of the “dark side” (now highly negative aspects), was introduced far before page 14: for example when describing the multifaceted nature of the human-animal relationship (section 2). We further mentioned the “dark side” and animal hoarding in the sections on empathy, on attachment and on anthropomorphism. Maybe these previous mentions of negative relationship, abuse and animal hoarding were missed by the reviewer.
Below we report previous sentences in the submitted manuscript referring to animal violence / abuse and animal hoarding:
Lines 86-90: “Personality has been associated with positive attitudes to animals and there is a relationship between psychopathic personality traits and animal abuse and violence [26, 35, 36]. Empathy and attachment are both related to the quality of human animal relationships, and problems in empathy, attachment and emotion regulation have been associated with animal abuse and cruelty [37- 40]”.
Lines 350-353: “There is also evidence that a lack of empathy is a characteristic of a psychological trait labelled as ‘callous and unemotional’ and an association between “callousness” and animal abuse during childhood and adolescence has been reported [173,174].”
Lines 379-385: “Finally, various studies showed an association between violence toward animals and a lack or suppression of animal-directed empathy [28,152], with empathy representing a mediating factor in aggression to both humans and animals [107]. Even though no mental disease has been specifically related to a lack of empathy towards animals, “hurting animals” is included in the diagnostic criteria of Conduct Disorder, and the last version of DSM (2013) includes a psychological disorder, Animal Hoarding, that has been related to impairment of empathy and attachment towards animals [11, 99,177, 178].”
Lines 495-498: “Notably, attachment difficulties with primary caregivers and attachment dysfunc-tions in adulthood are associated with cruelty and abuse towards animals [108, 209], and several studies show that in animal hoarding, animal suffering and neglect may occur in conjunction with a strong distorted attachment to animals [11, 213].”
Lines 661-664: “Although the motives underlying animal cruelty and abuse are multiple, complex [250,251] and do not make specific reference to anthropomorphism, they may be related to the need to control animals, prejudices against a particular species or breed or just dislike for a particular animal [252]. Some studies also found a significant association between anthropomorphism, hoarding behaviors and emotional attachment to possessions [234, 253, 254] and there is some evidence (e.g., [235]) that animal hoarders tend to anthropomorphize animals to a greater extent compared to non-hoarder animal owners.”
Rev1. This is a nice review of the literature but, in my view, does not make a significant contribution to the literature. There are other reviews available.
Response. We thank the reviewer for the positive comment. We disagree with the view that this work does not contribute to the literature. To our knowledge, although other reviews are available (some have been cited in the current work), this review considers not either the potential benefits or the existing “dark sides” of the human-animal relationship but focuses on the “dimensional” and “nuanced” nature of this relationship. We could not find reviews that concurrently addressed in some detail the variable nature of the human-animal relationship in relation to all the three human psychological aspects which are relevant for both interpersonal and intraspecific relationships. In case we missed the reviews mentioned by the Reviewer, we would appreciate having references for them, to consider them in our work.
Rev1. The frame using the three psychological processes mentioned is not particularly informative; nor is its application to hoarding.
Response. Unfortunately these statements are not very helpful in allowing us to improve our work. Thus, we were unable to figure out what the reviewer intended by “not very informative” and “not applicable to hoarding” unless he/she provides some clarification. As we reported above empathy, attachment and anthropomorphism are widely recognized as three key psychological processes affecting and modulating relationships between people and nonhuman animals; their characteristics and their functioning are object of growing interest in all the different areas of HARs. Thus, using them not “to frame the literature”, but to summarize how they are deeply involved in the complex nature of human animal relationship is something that can be informative.
Regarding Animal Hoarding, the involvement of empathy, attachment and anthropomorphism is a topic of interest for both researchers and clinicians. In addition, the characteristics and functioning of these processes in animal hoarders have not been explored in depth, and only rarely they have been compared in “normal” human-animal relationships and in hoarding situations. These three processes, although dysfunctional and compromised, play a role in animal hoarding, facilitating the onset of the phenomenon and determining the difference between clinical hoarding and just keeping many animals. Therefore, it seems to us that describing and discussing them is pertinent to animal hoarding.
Rev1. “…greatly reducing the review of the benefits of HAR literature and either reducing the implied claim that the three processes constructively reframe the literature on this subject or more convincingly showing that it does.
Response. Our aim was not to review the benefits of HAR and in the submitted paper 10 lines were dedicated to reporting some of this literature (Lines 157-166). However, to address the request we have now left just one sentence with a few references (Lines 159-164).
We do not claim that these three processes “reframe the literature”, and our goal in this review is less ambitious of what the reviewer states. What we aimed to do was just to highlight how these three human processes are pervasive and involved in both positive, poor and negative human-animal bonds and how some normal mechanisms involved in human-animal bonding are present, but compromised, in the phenomenon of animal hoarding. In any case, given the reviewer had this impression, when revising the manuscript, we have tried to make this clearer in the different sections of the manuscript, and in the Conclusions and afterthought we have tried to further clarify these aspects also adding an introductory sentence (Lines 955-960). We hope to have been successful in doing so.

Reviewer 2 Report
Small grammar, punctuation, and tense errors throughout (not enough to impede understanding, but may cause readers to have to read the sentence multiple times, for example), suggest careful proofreading
There is a fair bit of repetition of ideas and information throughout (a few examples in the detailed comments below). Suggest editing for repetition to strengthen paper.
Line 110: “… about their pet welfare but for…” Suggest either “care about their pet’s welfare” (specific) or “care about pet welfare” (general, which flows better with the following premise about farm animal welfare.
Line 113-114: The sentence beginning “As regards the animals' characteristics…” is a bit confusing. Suggest rewording for clarity: “Regarding animal characteristics, people generally do not see all animals as equal, as their physical and behavioral…”
Line 125: “Belief in animal mind appear to be…” Suggest “Belief in the animal mind appears to be…”
Line 128: “The propensity to use of animals is greater…” reword to “The propensity to use animals…”
Line 148: “…disagree on what a proper way to treat animals or a fair human-animal relationships is.” Suggest rewording sentence to “…disagree on the proper way to treat animals or a what a fair human-animal relationship is.”
Line 152: Authors state that Mota-Rojas et al. discussed effects of anthropomorphism, please add a brief sentence or two about Mota-Rojas et al.’s conclusions to strengthen their own arguments.
Line 162: be consistent in reference to human-animal relationship (authors used people-animal relationship here)
Line 173-174: First sentence of paragraph is unclear; are authors suggesting that in interpersonal relationships, humans have self-interest as well as caring about the other in the relationship? And, that this balance of self-interest and care for others is reflected in the human-animal relationship?
Line 192-193: authors write “literally by providing them opportunities to do things that they couldn’t otherwise do.” Please provide an example to strengthen your point.
Line 250: reword to “…consists of the affective…”
Lines 268-279: some repetition in these three paragraphs. Suggest making them a single paragraph, and deleting the sentence “Experience, education, and culture influence both affective and cognitive components of empathy.”
Line 317-321: this is repeated information (presented at lines 80-83)
Line 398: suggest replacing “mental disease” with “psychiatric diagnosis” or “psychological disorder”
Line 452: suggest rewording sentence to “…an individual views themself as lovable…” to avoid a binary approach to gender here
Line 571: provide a reference here (development of empathy for hunted animals)
Line 599-600: last part of final sentence in paragraph is a bit confusing, reword for clarity
Line 668: awkward wording, suggest reword first sentence of paragraph something like “People view and treat nonhuman animals in line with…”
Line 760: statement implies that animal hoarding is a significant problem in Brazil; did authors intend to state that the breadth of studies indicates that animal hoarding is a problem across the world?
Line 782: please clarify what is meant by “even breeder” in this context
Line 795: please provide reference for ‘overwhelmed caregivers’
Line 802: please provide reference for ‘rescue hoarders’
Line 813: please provide reference for ‘exploiter hoarders’
Line 875: missing parenthesis “…laboratory)”
Line 915: companion animal is hyphenated here but not elsewhere in the manuscript, edit for consistency of terms
Lines 931-932: awkward sentence, please reword for clarity
Line 990: typo emotional (vs emotionl as in text)
Line 1011: typo depending (vs deending in text)
Author Response
Response to Reviewer 2
We thank the reviewer for appreciating our manuscript and we have considered the proposed suggestions/comments.
Rev2. Small grammar, punctuation, and tense errors throughout (not enough to impede understanding, but may cause readers to have to read the sentence multiple times, for example), suggest careful proofreading
Response. We have checked the manuscript for grammar, punctuation, and tense errors.
Rev 2. There is a fair bit of repetition of ideas and information throughout (a few examples in the detailed comments below). Suggest editing for repetition to strengthen paper.
Response. We have revised the paper to minimize possible repetitions as suggested. However, some repetitions server to specify specific issues discussed in different sections of the paper (i.e., empathy, attachment or anthropomorphism) and were retained for that reason. For example, gender differences were repeated in the text because they are typical of both empathy (toward humans and animals) and attachment toward animals.
Rev 2. Line 110: “… about their pet welfare but for…” Suggest either “care about their pet’s welfare” (specific) or “care about pet welfare” (general, which flows better with the following premise about farm animal welfare.
Response. We have modified the text as suggested (Line 109).
Rev2. Line 113-114: The sentence beginning “As regards the animals' characteristics…” is a bit confusing. Suggest rewording for clarity: “Regarding animal characteristics, people generally do not see all animals as equal, as their physical and behavioral…”
Response. We have modified the text as suggested (Lines 112-113).
Rev 2. Line 125: “Belief in animal mind appear to be…” Suggest “Belief in the animal mind appears to be…”
Response. Done (Line125).
Rev 2. Line 128: “The propensity to use of animals is greater…” reword to “The propensity to use animals…”
Response. Done (Line 128).
Rev2. Line 148: “…disagree on what a proper way to treat animals or a fair human-animal relationships is.” Suggest rewording sentence to “…disagree on the proper way to treat animals or a what a fair human-animal relationship is.”
Response. The sentence has been reworded (Lines 148-149)
Rev2. Line 152: Authors state that Mota-Rojas et al. discussed effects of anthropomorphism, please add a brief sentence or two about Mota-Rojas et al.’s conclusions to strengthen their own arguments.
Response.The sentence has been slightly shortened and conclusion were added. The actual sentence is now: “For example, Mota-Rojas et al., [69] outlined how adverse consequences on pets’ welfare might depend on widespread and apparently affectionate and caring behaviors, such as dressing pets, application of cosmetics, letting them sleep in beds or overfeeding them, emphasizing that people’s behavior towards companion animals should be based on the understanding and respect of their natural needs rather than on supposed simi-larities and an affective involvement.” (Lines 150-155).
Rev2. Line 162: be consistent in reference to human-animal relationship (authors used people-animal relationship here).
Response. Thanks, it was a typo and has been corrected and we also checked throughout the text for possible errors.
Rev2. Line 173-174: First sentence of paragraph is unclear; are authors suggesting that in interpersonal relationships, humans have self-interest as well as caring about the other in the relationship? And, that this balance of self-interest and care for others is reflected in the human-animal relationship?
Response. Yes, this is what we suggest (now lines 165-) We intend that humans use the same psychological processes and mechanisms in interacting with conspecifics and with nonhuman animals. This point has been outlined again in the sections on empathy, attachment, and anthropomorphism, which initially summarize the features of these psychological characteristics in interpersonal relationships and then in interspecific relationships)
Rev2. Line 192-193: authors write “literally by providing them opportunities to do things that they couldn’t otherwise do.” Please provide an example to strengthen your point.
Response. As required we provided an example (Lines 185-187).
Rev2. Line 250: reword to “…consists of the affective…”
Response. Done (Lines 242)
Rev2. Lines 268-279: some repetition in these three paragraphs. Suggest making them a single paragraph, and deleting the sentence “Experience, education, and culture influence both affective and cognitive components of empathy.”
Response. This part has been rearranged to reduce repetitions (Lines 261-267)
Rev2. Line 317-321: this is repeated information (presented at lines 80-83).
Response. The section on empathy has been slightly modified and we have tried to limit repetitions, keeping the information that there is a gender effect in both empathy towards toward people and animals (Lines 305-310)
Rev2. Line 398: suggest replacing “mental disease” with “psychiatric diagnosis” or “psychological disorder”
Response. We agree and the term “psychological disorder” has been used throughout the text
Rev2. Line 452: suggest rewording sentence to “…an individual views themself as lovable…” to avoid a binary approach to gender here.
Response. We have changed the sentence to “individuals view themselves as lovable and…..” (Lines 435-436)
Rev2. Line 571: provide a reference here (development of empathy for hunted animals)
Response. Done (Lines 552-553).
Rev2. Line 599-600: last part of final sentence in paragraph is a bit confusing, reword for clarity
Response. The sentence has been reworded (Lines 582-585)
Rev2. Line 668: awkward wording, suggest reword first sentence of paragraph something like “People view and treat nonhuman animals in line with…”
Response. The sentence has been reworded (Line 653).
Rev2. Line 760: statement implies that animal hoarding is a significant problem in Brazil; did authors intend to state that the breadth of studies indicates that animal hoarding is a problem across the world?
Response. We intended to say that the available literature indicates that animal hoarding is a problem in several countries not just in Brazil. We have now clarified this point which was unclear (Lines 743-746).
Rev2. Line 782: please clarify what is meant by “even breeder” in this context
Response. We have clarified that also breeders can be involved in animal hoarding “but may also be breeders who initially bred animals for economic reasons”. (Lines 766-767).
Rev2. Line 795: please provide reference for ‘overwhelmed caregivers’
Response. References were reported at the start of the paragraph. However, we have now added them also for ‘overwhelmed caregivers’ (Line 780).
Rev2. Line 802: please provide reference for ‘rescue hoarders’
Response. Same as above (Line 787)
Rev2. Line 813: please provide reference for ‘exploiter hoarders’
Response. Same as above (Line 798).
Rev2. Line 875: missing parenthesis “…laboratory)”
Response. Thank you, it has been corrected
Rev2. Line 915: companion animal is hyphenated here but not elsewhere in the manuscript, edit for consistency of terms
Response. Thank you for noticing, it has been corrected and checked throughout the text.
Rev2. Lines 931-932: awkward sentence, please reword for clarity
Response. The sentence has been reworded (Lines 916-918).
Rev 2. Line 990: typo emotional (vs emotionl as in text)
Response. Thank you for noticing it, it has been corrected

Round 2
Reviewer 1 Report
I still find the paper of minimal contribution but am recommending its publication as the topic is so important and timely.